# Changes in Locomotor Activity Observed During Acute Nicotine Withdrawal Can Be Attenuated by Ghrelin and GHRP-6 in Rats

**DOI:** 10.3390/biomedicines13010143

**Published:** 2025-01-09

**Authors:** Jázmin Ayman, András Buzás, Roberta Dochnal, Miklós Palotai, Miklós Jászberényi, Zsolt Bagosi

**Affiliations:** 1Department of Obstetrics and Gynecology, Albert Szent-Györgyi Medical School, University of Szeged, 6725 Szeged, Hungary; aymanjazmin@gmail.com; 2Department of Surgery, Albert Szent-Györgyi Medical School, University of Szeged, 6725 Szeged, Hungary; buzas.andras@med.u-szeged.hu; 3Department of Pediatrics and Pedriatic Health Center, Albert Szent-Györgyi Medical School, University of Szeged, 6725 Szeged, Hungary; rdochnal@gmail.com; 4Department of Radiology, Brigham and Women’s Hospital, Harvard Medical School, Boston, MA 02115, USA; palotai@bwh.harvard.edu; 5Department of Pathophysiology, Albert Szent-Györgyi Medical School, University of Szeged, Szőkefalvi-Nagy Béla str. 6., 6720 Szeged, Hungary; jaszberenyi.miklos@med.u-szeged.hu

**Keywords:** nicotine, ghrelin, GHRP-6, dopamine, rats

## Abstract

**Background/Objectives:** Ghrelin and growth hormone-releasing peptide 6 (GHRP-6) are peptides which can stimulate GH release, acting through the same receptor. Ghrelin and its receptor have been involved in reward sensation and addiction induced by natural and artificial drugs, including nicotine. The present study aimed to investigate the impacts of ghrelin and GHRP-6 on the horizontal and vertical activity in rats exposed to chronic nicotine treatment followed by acute nicotine withdrawal. **Methods:** Male and female Wistar rats were exposed daily to intraperitoneal (ip) injection with 2 mg/kg nicotine or saline solution for 7 days, twice a day (at 8:00 and at 20:00). In parallel, the rats were exposed daily to an intracerebroventricular (icv) injection with 1 μg/2 μL ghrelin or 1 μg/2 μL GHRP-6 or saline solution for 7 days, once a day (at 8:00). On the morning of the eighth day (12 h after the last ip administration) and the ninth day (24 h after the last ip administration), the horizontal and vertical activity were monitored in a conducta system. **Results:** On the eighth day, in nicotine-treated rats a significant hyperactivity was observed, that was reduced significantly by ghrelin and GHRP-6. On the ninth day, in nicotine-treated rats a significant hypoactivity was assessed that was reversed significantly by ghrelin and GHRP-6. **Conclusions:** Based on the present results, the changes in horizontal and vertical activity observed after 12 and 24 h of nicotine withdrawal can be attenuated by ghrelin and GHRP-6.

## 1. Introduction

Nicotine is the main psychoactive component in tobacco that is responsible for the reward sensation and addiction produced by smoking [1]. During smoking, nicotine is rapidly absorbed from the lung alveoli into the systemic circulation; it passes the blood–brain barrier and reaches the midbrain, where it activates the nicotinic acetylcholine receptors (nAchRs) and stimulates the release of dopamine causing reward sensation and addiction [2,3]. The nAchR can be inhibited by mecamylamine, a non-selective and non-competitive nAchR antagonist that is usually administered orally in humans and subcutaneously (sc) or intracerebroventricularly (icv) in rodents, precipitating the symptoms of nicotine withdrawal [4,5,6] (Figure 1).

The nAchRs can be classified into two major subtypes, the muscle-type nAchRs, found in the neuromuscular junctions, and the neuronal-type nAchRs, shown on the neuronal bodies and nerve terminals [7]. Nicotine exerts its psychoactive effects via the neuronal-type nAchRs which are ligand-gated ion channels that normally respond to acetycholine and allow natrium or calcium ions to enter the neurons. Additionally, activation of the nAchR, located both pre- and postsynaptically, can influence the release of other neurotransmitters, including dopamine, glutamate and γ-aminobutyric acid (GABA) [7]. As regards dopamine, there are five subtypes of dopamine receptors grouped in two types with different functions (D1-type and D2-type) and two major dopaminergic pathways in the midbrain, the nigrostriatal and the mesolimbic pathway [8,9]. The nigrostriatal pathway arises from the dopaminergic neurons situated in the substantia nigra (SN) and terminates in the putamen and nucleus caudatus (caudate–putamen, CP), which constitute the dorsal striatum, modulating motor behavior and posture and contributing to the learning of motor programs and habits [8,9]. The mesolimbic pathway emerges from the dopaminergic neurons situated in the ventral tegmental area (VTA) and ends in the nucleus accumbens (NAcc) that represents the ventral striatum, but these neurons also send dopaminergic projections to the amygdala and prefrontal cortex, mediating reward sensation, emotion and motivation [8,9].

Ghrelin is a natural orexigenic peptide that can stimulate the growth hormone (GH) release [10,11]. Originally isolated from the rat stomach, ghrelin is released from the walls of the empty stomach during hunger, from where it is absorbed into the blood and reaches the brain, more specifically the ghrelin receptor found in the nucleus arcuatus (Arc) of the hypothalamus [10,11]. Reaching the hypothalamus, ghrelin activates the central orexigenic mediators of food intake, neuropeptide Y (NPY) and agouti-related peptide (AgRP), and consequently, the central orexigenic regulators, orexin and melanin-concentrating hormone (MCH) to stimulate food intake [10,11] (Figure 2).

Growth hormone-releasing peptide 6 (GHRP-6) is a synthetic Met-enkephalin derivative that can also stimulate the GH release acting through the same receptor [12,13]. GHRP-6 is usually considered an antagonist of the ghrelin receptor, but it can also act as an agonist, stimulating the GH release and even food intake [12,13]. The ghrelin receptor, previously known as the growth hormone secretagogue receptor (GHSR), is a G protein-coupled receptor that binds GH secretagogues, such as ghrelin and GHRP-6 [14,15]. Classically, the GHSR was classified in two subtypes, GHSR1a and GHSR1b, with GHSR1a regulating food intake and with GHSR1b inhibiting the activity of GHSR1a [16]. GHSR1a can found both in the central nervous system (CNS) and the periphery [17]. Centrally, it is expressed in the Arc of the hypothalamus, but also the laterodorsal tegmental nucleus (LDT) of the brainstem that has connections with the mesolimbic pathway [14,15]. Therefore, ghrelin and its receptor have been involved in reward sensation and addiction induced by natural and artificial drugs, including nicotine, but the picture regarding the interaction between ghrelin and nicotine is not clear [3,18,19,20,21,22].

In the present study, we aimed to investigate the impacts of ghrelin and GHRP-6 on the horizontal and vertical activity in rats exposed to chronic nicotine treatment followed by acute nicotine withdrawal. In one of our previous studies we have already investigated the participation of corticotropin-releasing factor (CRF) receptors (CRF1 and CRF2) in the changes in locomotor activity and the those of striatal dopamine release in rats exposed to similar in vivo conditions [23]. In the previous experiments, only male Wistar rats were used, because at that time, we believed that the behavior of females would be affected by too many variables, including hormonal fluctuations associated with the female reproductive cycle [24,25]. However, a meta-analysis demonstrated that female rats were not more variable regarding behavioral, electrophysiological, neurochemical and histological measures at any stage of the estrous cycle, than male rats [26]. Therefore, in the present experiments both male and female rats were used, and the stage of reproductive cycle of female rats was not taken into consideration. Nevertheless, the effects of chronic nicotine treatment and acute nicotine withdrawal on locomotion could be influenced by many factors, including sex, strain, age and housing conditions [27,28,29,30,31]. In general, female animals are less sensitive to the effect of nicotine, but more sensitive to the impact of acute nicotine withdrawal, compared to males [24,26,27,29,32]. Furthermore, different strains of rats react differently to nicotine: when Long–Evans and Sprague Dawley male and female rats were compared, the horizontal activity was more enhanced in Long–Evans females, and the vertical activity was unchanged in Sprague Dawley males [27]. In addition, rats of adolescent age exhibit increased sensitivity to the positive, rewarding effects of nicotine and decreased sensitivity to the negative, aversive effects of nicotine withdrawal, which may contribute to the higher risk to develop nicotine addiction in adolescents, compared to adults [31,33]. Moreover, the effects of nicotine on locomotion seems to be influenced by the housing conditions, as well [28]: the horizontal and vertical activity increased in male rats exposed to chronic nicotine treatment when they were housed together, and decreased when they were housed individually [28]. In contrast, chronic nicotine treatment did not induce any change in the horizontal and vertical activity of female rats, compared to control rats [28]. The influence of housing conditions were manifested even more robustly during acute nicotine withdrawal, at least in males [28].

## 2. Materials and Methods

### 2.1. Animals

The male and female Wistar rats used were provided by Toxi-Coop, Toxicological Research Center Zrt., Budapest, Hungary (N = 97). The rats were of adolescent age (about 6–7 weeks), but sexually maturized, weighing 150–250 g upon arrival. Before the experiments, the rats were housed together and kept at a constant temperature on a standard illumination schedule with 12 h light and 12 h dark periods (lights on from 6:00). Commercial food and tap water were available ad libitum. To minimize the effects of non-specific stress the rats were handled daily. During the experiments, the rats were treated in accordance with the instructions of the Ethical Committee for the Protection of Animals in Research, University of Szeged, Hungary.

### 2.2. Surgery

The rats were implanted with a stainless steel Luer cannula (10 mm long), aimed at the right lateral cerebral ventricle (LCV) under anesthesia with 60 mg/kg pentobarbital sodium (Euthanasol, CEVA-Phylaxia, Budapest, Hungary). The stereotaxic coordinates were 0.2 mm posterior and 1.7 mm lateral to the bregma, 3.7 mm deep from the dural surface, according to the stereotaxic atlas of the rat brain [34]. Cannulas were secured to the skull with acrylate and dental cement (Spofa Dental Adhesor, Prague, Czech Republic). The rats were allowed to recover for 7 days before the actual experiments started. After the experiments, the rats were decapitated, and 10 μL of methylene blue (Reanal Ltd., Budapest, Hungary) at 1 g/100 mL was injected icv and then the position of the cannula was inspected visually. Animals without the dye (3 out of 100) in the LCV were excluded from the final statistical analysis. There was no animal loss following anesthesia or surgery.

### 2.3. Treatments

The rats were exposed daily to intraperitoneal (ip) injection with 2 mg/kg nicotine (Sigma-Aldrich Inc., St. Louis, MO, USA) or 0.9% saline solution (B. Braun Inc., Melsungen, Germany) for 7 days, twice a day (at 8:00 and at 20:00). This dose and schedule of administration of nicotine usually produce a plasma nicotine level in rats similar to that found in individuals who smoke 1–2 packs of cigarettes a day [35]. In parallel, the rats were also exposed daily to icv injection with 1 μg/2 μL ghrelin or 1 μg/2 μL GHRP-6 or 2 µL of 0.9% of saline solution for 7 days, once a day (at 8:00). The doses of ghrelin and GHRP-6 were based on our previous studies from which the most effective doses were chosen. On the mornings of the 8th day (12 h after the last ip administration) and the 9th day (24 h after the last ip administration), the horizontal and vertical activity were monitored in a conducta system.

### 2.4. Behavioral Tests

The conducta system (Experimetria Ltd., Budapest, Hungary) was based on the principle of the open-field test that was described in our previous studies [36,37]. The main apparatus was a square open-field black box with a side length of 60 cm, surrounded by a 40 cm high wall and enlightened by a 60 W light bulb that was situated 1 m above the arena floor of the box. The arena floor was divided into 36 (6 × 6) small squares. Each animal was carried to the experimental room in their home cage and placed in the center of the arena, with which they were familiarized for 5 min. Then, the horizontal and vertical activity of the rats was monitored using five-by-five rows of photocell beams and registered by a computer for 10 min each. The horizontal activity measured the overall activity and arousal, while the vertical activity was a measure of exploratory and stereotype behavior. The box was cleaned between sessions with 96% ethyl-alcohol (Reanal Ltd., Budapest, Hungary).

### 2.5. Statistical Analysis

Statistical analysis of the results was performed by analysis of variance (Prism 7 Statistics, GraphPad Inc., La Jolla, CA, USA). The differences between groups were determined by one-way ANOVA followed by Tukey’s post hoc test and the Kruskal–Wallis test, followed by the Dunn post hoc test, which were preceded by the Shapiro–Wilk normality test. A probability level (*p* value) of 0.05 or less was accepted as indicating a statistically significant difference. The *p* values for pairwise comparison between the groups are summarized in Table 1.

## 3. Results

On the eighth day (12 h after the last ip administration), in nicotine-treated rats a significant hyperactivity was observed (F(5,45) = 17.28; *p* < 0.001 for horizontal activity and (F(5,45) = 1.97; *p* = 0.024 for vertical activity), that was reduced significantly by ghrelin and GHRP-6 (Figure 3 and Figure 4). When the rats were separated into male and female groups, the horizontal activity increased more significantly in males (F(5,25) = 9.04; *p* = 0.001), than females (F(5,20) = 7.86; *p* = 0.008) and accordingly, the impacts of ghrelin and GRHP-6 were statistically significant only in males, not females (Figure 3).

In comparison, the vertical activity increased significantly in both males F(5,25) = 11.67; *p* = 0.004) and females F(5,20) = 10.872; *p* = 0.050), but the effects of ghrelin and GRHP-6 were statistically insignificant in both sexes (Figure 4).

On the ninth day (24 h after the last ip administration), in nicotine-treated rats a significant hypoactivity was assessed (F(5,45) = 19.11; *p* < 0.001 for horizontal activity and (F(5,45) = 4.94; *p* = 0.013 for vertical activity), that was reversed significantly by ghrelin and GHRP-6 (Figure 5 and Figure 6). This time, when the rats were separated in two groups, the horizontal activity decreased significantly in both males (F(5,25) = 7.94; *p* < 0.001) and females (F(5,20) = 12.05; *p* < 0.001) treated with nicotine, and subsequently, the effects of ghrelin and GHRP-6 were significant in both sexes (Figure 5).

In contrast, even if the vertical activity decreased more remarkably in males (F(5,25) = 2.81; *p* = 0.045) than females (F(5,20) = 2.48; *p* = 0.079), the effects of ghrelin and GHRP-6 were similarly insignificant in both males and females (Figure 6).

## 4. Discussion

On the eighth day, the horizontal and vertical activity increased in rats exposed to 7 days of nicotine treatment. This finding is concordant with previous studies in which locomotor hyperactivity was reported on the fourth, the eighth and the tenth day of chronic nicotine exposure [23,38,39]. The locomotor hyperactivity observed can be explained by the increase in the concentration of dopamine and the density of dopamine receptors (D1-type and D2-type) in the striatum, but also the supersensitivity of the midbrain dopamine receptors that develops usually after a few days in response to nicotine [23,40]. Acute administration of nicotine stimulates the release of dopamine in both the dorsal and ventral striatum and this can cause reward sensation and locomotor hyperactivity in rats, as it was indicated by both in vivo and in vitro studies [41,42]. For example, nicotine infused into the striatum increased the dopamine output and the locomotor activity in freely moving rats, which were reduced by administration of mecamylamine [23,43]. In addition, superfusion of nicotine to the striatum increased the dopamine release in rats [44,45], which was reduced by the administration of mecamylamine [43,46]. During acute nicotine exposure both the mesolimbic and nigrostriatal dopaminergic pathways are activated, but usually there is a higher dopamine release in the NAcc than the CP [47,48,49,50]. This can be explained by the different expression of dopamine receptors (D1-type versus D2-type) in the two subdivisions of striatum, exerting distinct, stimulatory and inhibitory effects on the dopamine release [38,51]. During chronic nicotine exposure, the dopamine output can be decreased or increased in the striatum leading to different behavioral outcome, depending on whether tolerance or sensitization to nicotine would develop [8,9]. Tolerance is more likely to occur due to continuous infusion of nicotine, while behavioral sensitization develops typically due to intermittent injection of nicotine, and the sum of these competing phenomena can be manifested as locomotor hyperactivity or hypoactivity [8,9]. Also, chronic stimulation of the nAchR can influence the release of other neurotransmitters, including acetylcholine, glutamate and GABA, with diverse impacts on locomotor activity [7].

On the ninth day, the horizontal and vertical activity decreased in rats exposed to 1 day of acute nicotine withdrawal. The nicotine withdrawal syndrome has a somatic component, characterized by locomotor hypoactivity, increased appetite and weight gain, and an affective component, represented by dysphoria, anxiety and depression [52]. The physical signs start promptly within a few hours and peak around 24 h following nicotine cessation [52]. The affective symptoms may start early, but can persist from days to months, resulting in chronic nicotine withdrawal characterized by craving and increased risk to relapse [52]. Nicotine addiction develops due to a combination of the rewarding, positive actions produced by nicotine, and the avoidance of the aversive, negative effects induced by nicotine withdrawal [7]. This is reflected by our previous and present findings, according to which reward deficit and locomotor hypoactivity appear following 24 h, but not 12 h, of nicotine withdrawal, along with the signs of anxiety and depression [23,53]. Nevertheless, in the present study a general hypoactivity was assessed, while in our previous study, only the vertical activity and the ventral striatal dopamine release were decreased, while the horizontal activity and the dorsal striatal dopamine release were still increased following 1 day of nicotine withdrawal [23]. This can be explained by the different abilities of the two major subtypes of nAchR expressed in the dorsal and ventral striatum (α5 versus α6 subunits) for tolerance or behavioral sensitization to nicotine that can also be affected by daily peptide injection [47,48].

The locomotor hyperactivity observed in nicotine-treated rats on the eighth day was reduced significantly by ghrelin and GHRP-6. Also, the locomotor hypoactivity assessed in nicotine-treated rats on the ninth day was reversed significantly by both the natural and synthetic peptide acting through ghrelin receptor. GHRP-6 is usually considered an antagonist of the ghrelin receptor, but it can also act as an agonist in many aspects, including the stimulation of GH release and food intake [12,13]; thus, the similar impacts of ghrelin and GHRP-6 on the locomotor effects of nicotine are not surprising and could be related to the neuroprotective effects of GH secretagogues that were demonstrated in the hypothalamus and cerebellum [54,55]. Therefore, the interaction between ghrelin, nicotine and GHRP-6 may occur in various brain regions, including the hypothalamus and cerebellum [56,57,58,59], but it is most probably mediated by the cholinergic–dopaminergic reward link, which encompasses the afferent cholinergic projection from the LDT to the VTA, and the mesolimbic dopaminergic pathway [1,60,61,62,63,64,65]. This observation is supported by in vivo studies in which ghrelin injected peripherally or directly into the VTA increased the dopamine release in the NAcc, locomotor activity and food consumption in rats [66,67,68,69], but also in vitro studies in which ghrelin administered locally produced a concomitant release of acetylcholine in the LDT and dopamine in the NAcc in rats [63,64,65,70,71,72]. The interaction between ghrelin and nicotine may also take place in the extended amygdala, a functional unit that includes anatomical regions, such as the central amygdala (CeA), the bed nucleus of stria terminalis (BNST) and the shell of the NAcc. This speculation is based on in vivo studies in which ghrelin and nicotine stimulated the dopamine release in the midbrain, including the amygdala and striatum [23,73,74,75], but also in vitro studies in which ghrelin and nicotine stimulated the dopamine release in the amygdala, BNST and striatum [76,77,78]. Moreover, when administered together, ghrelin amplified the nicotine-induced release of dopamine in the BNST and striatum, and this effect was reversed partly by mecamylamine and partly by GHRP-6 [76,78]. We presume that the excess dopamine in the ventral and dorsal striatum may promote the positive, rewarding actions produced by chronic nicotine administration, whereas the deficiency of dopamine in the central amygdala, the BNST and the shell of NAcc may mediate the negative, aversive symptoms induced by acute nicotine withdrawal [79,80].

In the present study, the rats were of the same strain, age and housing conditions, so the only factor that could influence the horizontal and vertical activity was sex. As regards horizontal activity, female rats were less sensitive to the locomotor effects of nicotine and this finding has been indicated by previous studies [24,25,26,81,82,83]. As regards vertical activity, female rats were more sensitive to the locomotor effects of nicotine and this finding also has been suggested by previous studies [27,28,29,31,84,85,86]. When the rats were separated in male and female groups, it was only the horizontal activity that was decreased significantly by nicotine withdrawal and was reversed significantly by ghrelin and GHRP-6 treatment in both sexes. The lack of statistically significant effect in the case of the other parameters of locomotion could be related to the relatively small sample size achieved after dividing the rats into two separate groups. In humans, women exhibit more rapid escalation from casual drug taking to drug addiction, express a wider range of drug withdrawal symptoms when drug taking stops, and tend to exhibit greater vulnerability than men in terms of treatment outcome [26,81]. In rodents, short-term estradiol intake in female rats enhances acquisition and escalation of drug taking, motivation for addictive drugs, and relapse-like behaviors, that can be explained by a sex difference in the dopamine response in the ventral and dorsal striatum [26,81]. For instance, estradiol treatment of ovariectomized female rats enhances the dopamine release in the dorsolateral striatum, but not in the NAcc, and when drug taking becomes habitual, dopamine release increases in the dorsolateral striatum and decreases in the NAcc [26,81]. Therefore, the sex difference in the balance between the two dopaminergic pathways projecting to the CP and NAcc may underlie the sex differences in addiction [26,81]. The sex differences in the CRF system regulating the hypothalamic–pituitary–adrenal (HPA)-axis and the locus coeruleus-norepinephrine (LC-NE) arousal system may explain why female rats are more sensitive to the effects of nicotine withdrawal that results in activation of both these systems [24,25]. For example, there are significant sex differences in CRF functions ranging from its presynaptic regulation to its postsynaptic efficacy, but also in CRF receptor expression, distribution, trafficking and signaling that results in increased reactivity to stress in females during drug withdrawal [24,25]. In addition, there are important sex differences in the structure and function of the LC-NE system and its projections, as estrogen administration increases the capacity for NE synthesis and decreases NE degradation, potentially increasing arousal in females during drug withdrawal [24,25].

In addition to the relatively small sample size of the male and female groups after separation, our study may also have other limitations [87]. For example, instead of repeated ip or sc injections, other animal models of nicotine dependence and withdrawal, such as continuous nicotine infusion via osmotic minipumps, oral nicotine intake (drinking), nicotine vapor exposure and tobacco smoke exposure, could be considered more appropriate regarding construct validity, face validity and predictive validity [87]. When using ip injections, at least 4 days of repeated nicotine injections are required to induce dependence in mice and rats [87]. The advantage of this method is that the dose and time of nicotine administration are well controlled and produce fluctuating plasma nicotine levels, similarly to a smoker who smokes cigarettes at certain intervals which leads to fluctuating plasma nicotine levels [87]. The disadvantage of this method is that the rate of nicotine absorption from ip or sc injections is slower compared to when nicotine is inhaled, and therefore nicotine injections might be less rewarding than the inhalation of nicotine [87]. In addition, repeated ip or icv injection of nicotine can lead to the accumulation of nicotine in the system producing toxicity [87]. Rats are less sensitive to the toxic effects of nicotine, but repeated injections with high doses of nicotine (6 mg/kg, sc; 3 injections per day for 7 days) can lead to a relatively high mortality rate, which is not observed when lower doses of nicotine (1–3 mg/kg sc, 2 injections per day for 7 days) are administered [87]. Repeated ip or sc injections may also be a source of stress, although this was demonstrated to be compensated by the daily handling of the animals [88]. The implantation of icv cannula by surgery and the repeated icv injections may be also stressful for the animals, however this was also partly ameliorated by the 7 days of recovery before the actual experiments had started. To accurately model human smoking, the blood nicotine and cotinine levels in rodents must be similar to those in smokers, but in the present study these were not measured. Also, we could not determine exactly the extent of the positive euphoric state and the negative dysphoric state that occurred in rats after 7 days of nicotine treatment and 1 day of acute nicotine withdrawal, respectively. Furthermore, the impacts of ghrelin and GHRP-6 on the locomotor activity were observed only after 7 days of nicotine treatment during spontaneous nicotine withdrawal in two time points and observation of the withdrawal symptoms earlier or later, and their precipitation with mecamylamine or comparison with the effects of drugs with therapeutic potential, such as bupropion and varenicline, were not studied. Since in the present study the rats were of the same strain, age and housing conditions, the impact of these factors on horizontal and vertical activity, and the effect of ghrelin and GHRP-6 when these conditions vary, was not determined either.

Nevertheless, one of our previous studies using male Wistar rats, and including both in vivo and in vitro experiments, indicated that different doses (0.5–5 µg) of ghrelin administered icv cause significant increases in both horizontal and vertical activity monitored by the conducta apparatus, while only the dose of 5 µg evokes a significant increase in spontaneous locomotor activity recorded by telemetry, which was associated with dose-dependent increases in plasma corticosterone concentration reflecting the activation of the HPA-axis. The locomotor hyperactivity observed was diminished by the non-selective CRF antagonist α-helical CRF(9-41) and the non-selective dopamine antagonist haloperidol, with higher affinity for D2 receptors, suggesting that both CRF release and dopaminergic neurotransmission are involved in the ghrelin-evoked locomotor responses [69]. Administration of GHRP-6 at 10 µg intravenously (iv) can also activate the HPA-axis resulting in small, but significant, increases in plasma concentrations of ACTH and corticosterone in rats [89]. As these GH secretagogues do not release ACTH directly, they probably interact with the hypothalamic neurohormones regulating ACTH release, such as CRF and arginine vasopressin (AVP), but, the ability of GHRP-6 to modulate the dopaminergic neurotransmission, just like ghrelin does, cannot be excluded [89]. Taken together, the present and previous studies suggest that ghrelin and GHRP-6 may modulate the acetylcholine release in the LDT and consequently, the release of dopamine in the NAcc and CP, compensating for the excess and deficiency of dopamine in periods of chronic nicotine treatment and acute nicotine withdrawal, respectively (Figure 7). In this order of thoughts, ghrelin receptor may represent a new target in the therapy of nicotine addiction.

## 5. Conclusions

Based on the present study, we conclude that the changes in horizontal and vertical activity observed after 12 h and 24 h of nicotine withdrawal can be attenuated by ghrelin and GHRP-6.

## Figures and Tables

**Figure 1 biomedicines-13-00143-f001:**
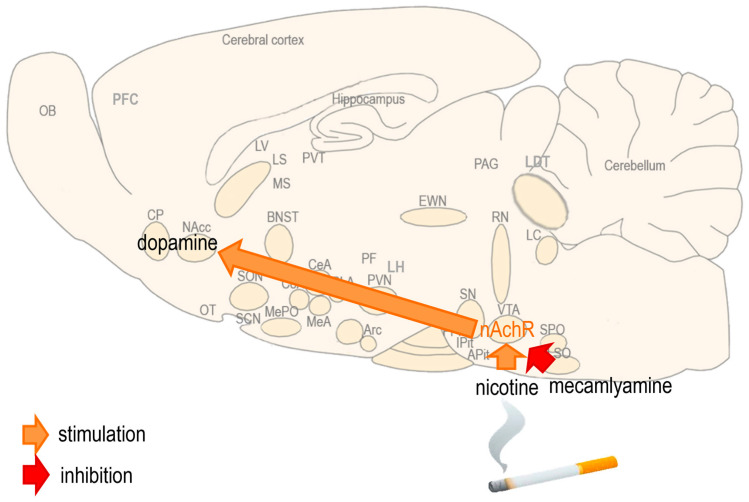
The actions of nicotine and mecamylamine on the brain. During smoking, nicotine reaches the brain and activates the nicotinergic acetylcholine receptor (nAchR) in the ventral tegmental area (VTA) and substantia nigra (SN), stimulating the release of dopamine in the nucleus accumbens (NAcc) and caudate–putamen (CP). Mecamylamine can inhibit the nAchR and precipitate the symptoms of nicotine withdrawal.

**Figure 2 biomedicines-13-00143-f002:**
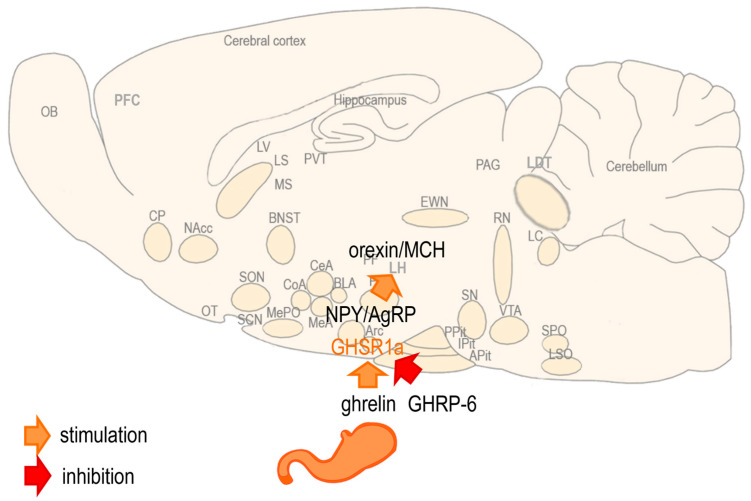
The actions of ghrelin and GHRP-6 on the brain. During hunger, ghrelin reaches the brain and activates the ghrelin receptor (GHSR1a) in the hypothalamus. In the hypothalamus, it activates the orexigenic mediators, neuropeptide Y (NPY) and agouti-related peptide (AgRP), expressed in the nucleus arcuatus (Arc), and consequently, the orexigenic regulators, orexin and melanin-concentrating hormone (MCH), expressed in the lateral hypothalamus, leading to increased food intake and body weight. Growth hormone-releasing peptide 6 (GHRP-6) is considered an antagonist of the GHSR1a, although it can act as an agonist stimulating the GH release and mimicking the orexigenic effects of ghrelin.

**Figure 3 biomedicines-13-00143-f003:**
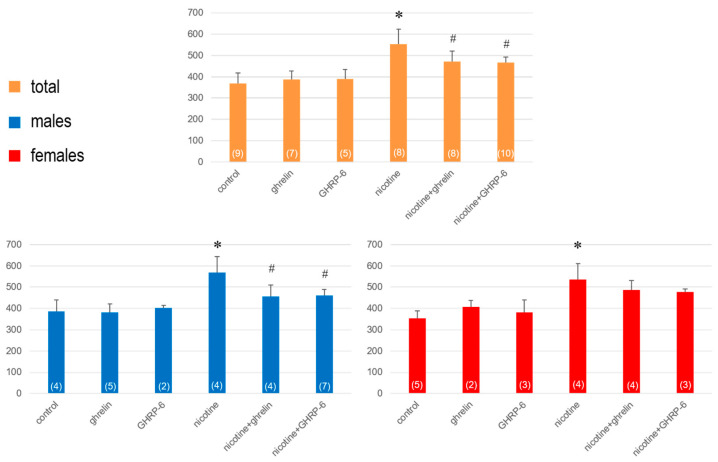
The horizontal activity determined on the 8th day in rats exposed to 7 days of nicotine treatment. Values are presented as means ± SEM. The numbers in parentheses represent the number of animals in each group. A statistically significant difference was accepted for *p* < 0.05 and indicated with * for nicotine ip + saline icv vs. saline ip + saline icv and with # for nicotine ip + ghrelin or GHRP-6 icv vs. nicotine ip + saline icv.

**Figure 4 biomedicines-13-00143-f004:**
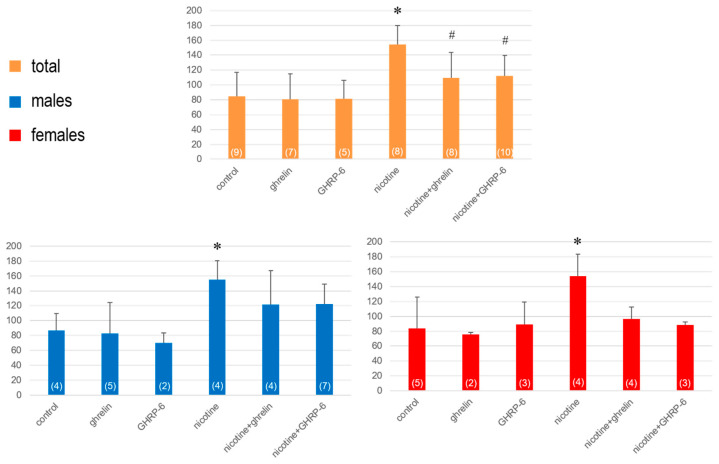
The vertical activity determined on the 8th day in rats exposed to 7 days of nicotine treatment. Values are presented as means ± SEM. The numbers in parentheses represent the number of animals in each group. A statistically significant difference was accepted for *p* < 0.05 and indicated with * for nicotine ip + saline icv vs. saline ip + saline icv and with # for nicotine ip + ghrelin or GHRP-6 icv vs. nicotine ip + saline icv.

**Figure 5 biomedicines-13-00143-f005:**
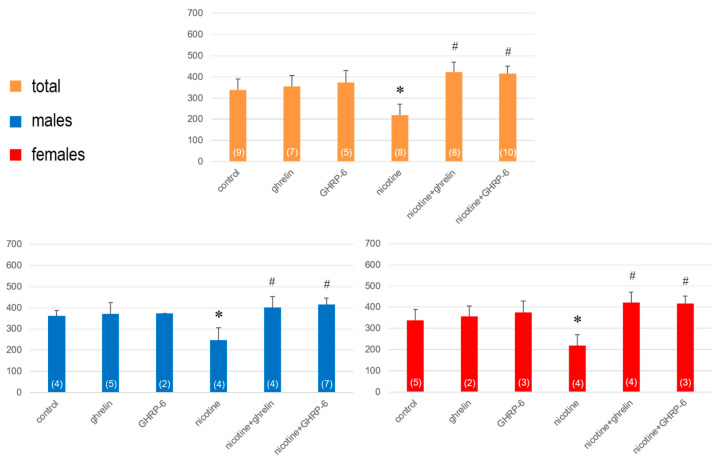
The horizontal activity determined on the 9th day in rats exposed to 7 days of nicotine treatment and 1 day of withdrawal. Values are presented as means ± SEM. The numbers in parentheses represent the number of animals in each group. A statistically significant difference was accepted for *p* < 0.05 and indicated with * for nicotine ip + saline icv vs. saline ip + saline icv and with # for nicotine ip + ghrelin or GHRP-6 icv vs. nicotine ip + saline icv.

**Figure 6 biomedicines-13-00143-f006:**
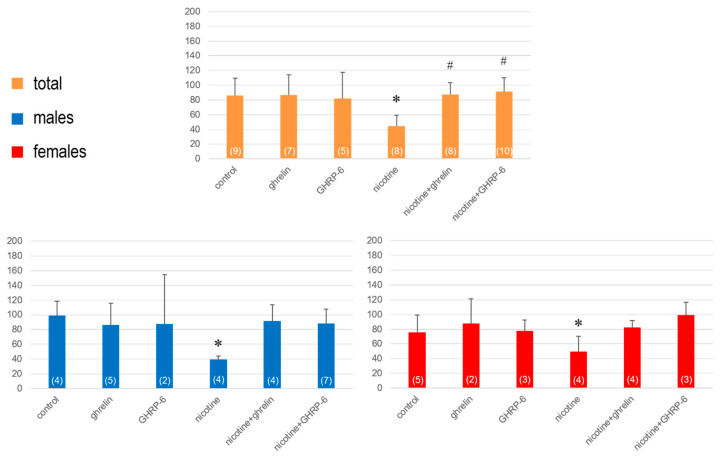
The vertical activity determined on the 9th day in rats exposed to 7 days of nicotine treatment and 1 day of withdrawal. Values are presented as means ± SEM. The numbers in parentheses represent the number of animals in each group. A statistically significant difference was accepted for *p* < 0.05 and indicated with * for nicotine ip + saline icv vs. saline ip + saline icv and with # for nicotine ip + ghrelin or GHRP-6 icv vs. nicotine ip + saline icv.

**Figure 7 biomedicines-13-00143-f007:**
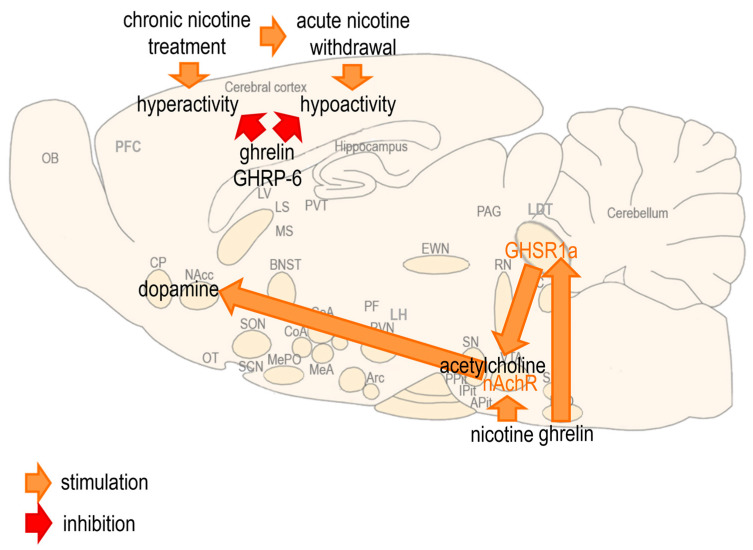
The interaction of ghrelin and nicotine in the brain. Chronic nicotine treatment is associated with general hyperactivity, whereas acute nicotine withdrawal with general hypoactivity can be attenuated by ghrelin and growth hormone-releasing peptide 6 (GHRP-6). Both ghrelin and nicotine can stimulate the dopamine release in the striatum, represented by nucleus accumbens (NAcc) and caudate–putamen (CP). The interaction between nicotine and ghrelin is most probably mediated by the cholinergic–dopaminergic reward link, which encompasses the afferent cholinergic projection from the laterodorsal tegmentum (LDT) to the ventral tegmental area (VTA), and the mesolimbic dopaminergic pathway that emerges from the dopaminergic neurons situated in the VTA and ends in the NAcc.

**Table 1 biomedicines-13-00143-t001:** The summary of the statistical analysis (F, female; M, male; *p*, probability value; T, total).

Parameter	*p* for Nicotine vs. Control	*p* for Nicotine + Ghrelin vs. Nicotine	*p* for Nicotine + GHRP-6 vs. Nicotine
Horizontal activity after 12 h	<0.001 (T) <0.001 (M) 0.001 (F)	0.016 (T) 0.031 (M) 0.742 (F)	0.006 (T) 0.019 (M) 0.679 (F)
Vertical activity after 12 h	0.001 (T) 0.007 (M) 0.002 (F)	0.019 (T) 0.133 (M) 0.062 (F)	0.026 (T) 0.476 (M) 0.342 (F)
Horizontal activity after 24 h	<0.001 (T) 0.016 (M) 0.002 (F)	<0.001 (T) 0.001 (M) <0.001 (F)	<0.001 (T) <0.001 (M) <0.001 (F)
Vertical activity after 24 h	0.006 (T) 0.036 (M) 0.038 (F)	0.006 (T) 0.084 (M) 0.231 (F)	0.001 (T) 0.075 (M) 0.876 (F)

## Data Availability

The datasets generated during the current study are available from the corresponding author upon reasonable request.

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
