# Peer review of "Changes in Locomotor Activity Observed During Acute Nicotine Withdrawal Can Be Attenuated by Ghrelin and GHRP-6 in Rats"

_biomedicines, 2025, doi:10.3390/biomedicines13010143_

Round 1
Reviewer 1 Report
Comments and Suggestions for Authors
The authors investigated the effects of ghrelin and growth hormone-releasing peptide 6 (GHRP-6) on horizontal and vertical activity in rats subjected to chronic nicotine treatment followed by acute withdrawal. They found that ghrelin and GHRP-6 reduce nicotine-induced hyperactivity at 12 hours and reverse hypoactivity caused by withdrawal at 24 hours, suggesting a role for dopamine. The study clearly outlines its objectives and research questions. The experimental design is well structured, with appropriate controls and sample sizes. The results are significant and contribute to our understanding of the neurobiological mechanisms underlying nicotine addiction and withdrawal. The paper is well-written, with clear and concise language. , I also suggest improving the manuscript by underlining the brain role of ghrelin and GHRP-6: 10.1016/j.brainresbull.2017.06.002 ;10.1111/j.1460-9568.2009.06770.x
While the study acknowledges sex differences in the effects of nicotine and the peptides, a deeper exploration of the underlying mechanisms could be beneficial. The authors could discuss the limitations of the study, such as the use of animal models and the potential impact of other factors on the results. The discussion could be further strengthened by providing a more nuanced interpretation of the sex differences observed. A more detailed description of the statistical analysis used could be provided.
With minor revisions, it can be further improved.
Author Response
Reviewer 1 wrote:
(our answers were written in red)
The authors investigated the effects of ghrelin and growth hormone-releasing peptide 6 (GHRP-6) on horizontal and vertical activity in rats subjected to chronic nicotine treatment followed by acute withdrawal. They found that ghrelin and GHRP-6 reduce nicotine-induced hyperactivity at 12 hours and reverse hypoactivity caused by withdrawal at 24 hours, suggesting a role for dopamine. The study clearly outlines its objectives and research questions. The experimental design is well structured, with appropriate controls and sample sizes. The results are significant and contribute to our understanding of the neurobiological mechanisms underlying nicotine addiction and withdrawal. The paper is well-written, with clear and concise language.
- I also suggest improving the manuscript by underlining the brain role of ghrelin and GHRP-6: 10.1016/j.brainresbull.2017.06.002; 10.1111/j.1460-9568.2009.06770.x
The references have been cited in the revised form of the manuscript, consequently the third paragraph of the Discussion section has been rewritten, as follows:
”GHRP-6 is usually considered an antagonist of the ghrelin receptor, but it can also act as an agonist in many aspects, including the stimulation of GH release and food intake, thus, the similar impacts of ghrelin and GHRP-6 on the locomotor effects of nicotine are not surprising and could be related to the neuroprotective effects of GH secretagogues that was demonstrated in the hypothalamus and cerebellum.”
- While the study acknowledges sex differences in the effects of nicotine and the peptides, a deeper exploration of the underlying mechanisms could be beneficial.
To improve our manuscript more explanations have been added regarding sex differences, consequently the fourth paragraph of the Discussion section has been rewritten, as well.
- The authors could discuss the limitations of the study, such as the use of animal models and the potential impact of other factors on the results.
To further improve our manuscript more explanations have been added regarding limitations of the study with a special focus on advantages and disadvantages of the ip and icv injections, consequently a fifth paragraph has been added to the Discussion section.
- The discussion could be further strengthened by providing a more nuanced interpretation of the sex differences observed.
As we mentioned before, more explanations have been added to the Discussion section of the revised manuscript including a more nuanced interpretation of sex differences in the CRF system regulating the HPA-axis and the LC-NE system mediating arousal that could be in the background of sex differences in nicotine addiction.
- A more detailed description of the statistical analysis used could be provided. With minor revisions, it can be further improved.
A more detailed description of the statistical analysis in the Results section, including the F distribution and degrees of freedom and p values has been provided in the revised manuscript when one-way ANOVA was performed. Additionally, there is table before the Results section summarizing the p values when post-hoc test was performed for pairwise comparison between the groups.
Thank you for the constructive critics, we hope that by answering them and including the corrections and some of our detailed answers into the revised form we could improve our manuscript!
Best wishes,
Jázmin Ayman, MD and Zsolt Bagosi, MD, PhD

Reviewer 2 Report
Comments and Suggestions for Authors
In this article, Jazmin et al. analyze experimental data on the effects of nicotine withdrawal and the potential regulatory role of ghrelin and GHRP-6 on these effects. Their results provided new insights into the neurobiological mechanisms of nicotine addiction and withdrawal, and new targets for the treatment of nicotine addiction. Before considering publishing in our journal, some revisions are needed.
Comments:
1. The title of the manuscript is too long, not concise enough, please revise
2. The abbreviation format for the “xx day” in the article is not consistent, such as lines 28, 30, and 31 in the Abstract. Also, check the abbreviations for the other days in the text and suggest standardizing the format.
3. In line 44, "mecamylamine" is an orally administered, non selective, and non competitive nAChR(Acetylcholine receptor antagonists). The description of it in lines 44-45 does not indicate that it is an oral medication, and in Figure 1, "mecamylamine" is labeled in the brain, which easily leads people to think that it is an nAChR antagonist produced in the brain. Therefore, it is necessary to indicate that it is an oral medication to avoid such misunderstandings; And in the figure, its labeling should also be labeled outside the brain instead of inside, just like how nicotine is produced by smoke.
4. There is a phenomenon of excessive use of hyphens in the text. It is recommended to check whether the use of hyphens is appropriate throughout the entire text. For example, there are many places such as "do-pamine" on line 50, "ac-activation" on line 57, "re-lease" on line 72, and "female ani-mals" on line 117, which will not be listed here. Even in line 81, "GHS-R1a" and in line 86, "GHSR1a" should be the same substance as "GHSR-1a" in Figure 2, but due to the misuse of hyphens, three different representations have emerged.
5. In lines 85 and 88, "Growth hormone releasing peptide 6 (GHRP-6)" does not need to be consistently represented by its full name and abbreviation. It only needs to be represented by its full name and abbreviation when it first appears, and can be directly represented by its abbreviation afterwards. If the same situation exists elsewhere in the text, the same applies.
6. Please explain the meaning of the numbers in parentheses in Figures 3-6.
7. Please discuss how ghrelin and GHRP-6 affect the behavior of rats? What is the mechanism?
Author Response
Reviewer 2 wrote:
(our answers were written in red)
In this article, Jazmin et al. analyze experimental data on the effects of nicotine withdrawal and the potential regulatory role of ghrelin and GHRP-6 on these effects. Their results provided new insights into the neurobiological mechanisms of nicotine addiction and withdrawal, and new targets for the treatment of nicotine addiction. Before considering publishing in our journal, some revisions are needed.
- The title of the manuscript is too long, not concise enough, please revise.
The Title has been shortened according to the Reviewer’s request. The revised manuscript is entitled: ”Changes in locomotor activity observed during acute nicotine withdrawal can be attenuated by ghrelin and GHRP-6 in rats”
- The abbreviation format for the “xx day” in the article is not consistent, such as lines 28, 30, and 31 in the Abstract. Also, check the abbreviations for the other days in the text and suggest standardizing the format.
The Reviewer is right, therefore the abbreviations for the days was standardized in the revised manuscript, using superscripts consistently.
- In line 44, "mecamylamine" is an orally administered, non selective, and non competitive nAChR(Acetylcholine receptor antagonists). The description of it in lines 44-45 does not indicate that it is an oral medication, and in Figure 1, "mecamylamine" is labeled in the brain, which easily leads people to think that it is an nAChR antagonist produced in the brain. Therefore, it is necessary to indicate that it is an oral medication to avoid such misunderstandings; And in the figure, its labeling should also be labeled outside the brain instead of inside, just like how nicotine is produced by smoke.
The Reviewer is right, therefore an additional sentence has been added regarding the possible routes of administration of mecamylamine in humans and rats and the all figures have been changes, where synthetic drugs were labeled on the brain.
- There is a phenomenon of excessive use of hyphens in the text. It is recommended to check whether the use of hyphens is appropriate throughout the entire text. For example, there are many places such as "do-pamine" on line 50, "ac-activation" on line 57, "re-lease" on line 72, and "female ani-mals" on line 117, which will not be listed here. Even in line 81, "GHS-R1a" and in line 86, "GHSR1a" should be the same substance as "GHSR-1a" in Figure 2, but due to the misuse of hyphens, three different representations have emerged.
Indeed, there were too many hyphens in the text, that was because the text was copy-pasted from an 2023 template into the 2024 version. But the revised manuscript was put in the 2025 template directly, therefore such mistakes will be avoided. GHSR1a has been also used consistently in the revised manuscript, the Reviewer was perfectly right about the three different represantions.
- In lines 85 and 88, "Growth hormone releasing peptide 6 (GHRP-6)" does not need to be consistently represented by its full name and abbreviation. It only needs to be represented by its full name and abbreviation when it first appears, and can be directly represented by its abbreviation afterwards. If the same situation exists elsewhere in the text, the same applies.
Indeed, every abbreviation should be mentioned just once, the first time it appears in the Abstract and then in the body of the article, we have tried to comply to that rule when rewriting the article. But the Legends of Figures still contain some abbreviations, including GHRP-6, that is explained for a better understanding of the Figure itself.
- Please explain the meaning of the numbers in parentheses in Figures 3-6.
The explanation was missing, indeed, therefore, we have added that The numbers in parentheses represent the number of animals in each group to the Legend of the Figures mentioned.
- Please discuss how ghrelin and GHRP-6 affect the behavior of rats? What is the mechanism?
Based on previous studies, besides modulation of the dopaminergic neurotransmission, the mechanism could be related to CRF regulating the HPA axis and LC-NE arousal system. An additional paragraph has been added to the Discussion section of the revised manuscript to detail that.
Thank you for the constructive critics, we hope that by answering them and including the corrections and some of our detailed answers into the revised form we could improve our manuscript!
Best wishes,
Jázmin Ayman, MD and Zsolt Bagosi, MD, PhD

Reviewer 3 Report
Comments and Suggestions for Authors
The authors addresses an important and novel topic, exploring the attenuation of nicotine withdrawal effects by ghrelin and GHRP-6 in rats. While the study demonstrates rigorous methodology and relevant findings, its presentation, analysis, and discussion need refinement to enhance clarity and impact. Here are some comments:
1. The authors only display the results of day 8 and day 9. What the results before day 8 and after day9? Investigate the long-term effects of ghrelin and GHRP-6 on nicotine withdrawal behaviors.
2. Explore the interaction between ghrelin signaling and other neurotransmitter systems beyond dopamine.
3. Examine dose-response relationships for both ghrelin and GHRP-6 in mitigating withdrawal symptoms.
4. Correct typographical errors in line 29, 166,175 “monitorized” → “monitored”;
5. line 193, 201 what the meaning of term "GRPP-6" ? is the typo error?
Author Response
Reviewer 3 wrote:
(our answers were written in red)
The authors addresses an important and novel topic, exploring the attenuation of nicotine withdrawal effects by ghrelin and GHRP-6 in rats. While the study demonstrates rigorous methodology and relevant findings, its presentation, analysis, and discussion need refinement to enhance clarity and impact. Here are some comments:
- The authors only display the results of day 8 and day 9. What the results before day 8 and after day9? Investigate the long-term effects of ghrelin and GHRP-6 on nicotine withdrawal behaviors.
Indeed, we only displayed the rsults of day 8 and 9, because the behavioral tests were performed only in these two time points. We have added an additional paragraph to the Discussion section of the revised manuscript regarding limitations of our study, including that issue.
- Explore the interaction between ghrelin signaling and other neurotransmitter systems beyond dopamine.
Based on previous studies, we have explored the interaction between ghrelin and CRF, AVP and norepinephrine, and added an additional paragraph to the Discussion section regarding that issue.
- Examine dose-response relationships for both ghrelin and GHRP-6 in mitigating withdrawal symptoms.
“The doses of ghrelin and GHRP-6 were based on our previous studies from which the most effective doses were chosen.” This has been mentioned in the Materials and Methods section of the revised manuscript.
- Correct typographical errors in line 29, 166,175 “monitorized” → “monitored”;
The Reviewer is right, monitorized is not correct, the word monitored was used instead consistently all along the revised mansucript.
- line 193, 201 what the meaning of term "GRPP-6" ? is the typo error?
The Reviewer is right again, GRPP-6 is a typo error, the abbreviation GHRP-6 was used instead consistently all along the revised mansucript.
Thank you for the constructive critics, we hope that by answering them and including the corrections and some of our detailed answers into the revised form we could improve our manuscript!
Best wishes,
Jázmin Ayman, MD and Zsolt Bagosi, MD, PhD

Round 2
Reviewer 2 Report
Comments and Suggestions for Authors
I have checked the revised version of manuscript. I think this manuscript can be acceptable in current form. No additional comments.
Reviewer 3 Report
Comments and Suggestions for Authors
The authors now have successfully addressed all my concerns.